# Status of Inappropriate Complementary Feeding and Its Associated Factors Among Infants of 9–23 Months

**DOI:** 10.3390/nu16244379

**Published:** 2024-12-19

**Authors:** Iqra Ashraf, Prince L. Bestman, Abdullah A. Assiri, Ghulam Mustafa Kamal, Jalal Uddin, Jiayou Luo, Khalid M. Orayj, Azfar A. Ishaqui

**Affiliations:** 1Department of Maternal and Child Health, Xiangya School of Public Health, Central South University, Changsha 410083, China; 216919002@csu.edu.cn (I.A.); plbestman@csu.edu.cn (P.L.B.); 2Department of Clinical Pharmacy, College of Pharmacy, King Khalid University, Abha 61441, Saudi Arabia; aalabdullah@kku.edu.sa (A.A.A.); korayg@kku.edu.sa (K.M.O.); amianishaqui@kku.edu.sa (A.A.I.); 3Institute of Chemistry, Khwaja Fareed University of Engineering & Information Technology, Rahim Yar Khan 64200, Pakistan; mustafa.kamal@kfueit.edu.pk; 4Department of Pharmaceutical Chemistry, College of Pharmacy, King Khalid University, Abha 61441, Saudi Arabia

**Keywords:** inappropriate complementary feeding, associated factors, minimum meal frequency, minimum dietary diversity, minimum acceptable diet

## Abstract

**Background:** Inappropriate complementary feeding during the first two years of life significantly impacts children’s health, increasing risks of malnutrition and illness. **Methods**: This study investigates factors influencing early feeding patterns among 600 mothers of children aged 9–23 months in selected hospitals in Punjab, Pakistan. Using a structured questionnaire, data were collected and analyzed, with associations measured by odds ratios (ORs) and 95% confidence intervals (CIs). **Results:** The results showed the key indicators of inappropriate complementary feeding among young children, including timely complementary feeding, minimum meal frequency, dietary diversity, and acceptable diet. The rates for these factors were found to be 60.3%, 32.7%, 24.6%, and 48.5%, respectively. The study identified several significant factors influencing these practices. Key predictors of inappropriate feeding included the order of birth, the mother’s employment status, parental education, the number of children, household income, maternal knowledge, and maternal health. **Conclusion:** The findings underscore that maternal education, employment, and health significantly influence complementary feeding. Targeted interventions and education programs are essential to support healthy feeding behaviors, especially for mothers facing challenges related to education, work, or health conditions. Addressing these practices can improve child health outcomes, contributing to economic growth and a healthier future for Pakistan’s youngest population.

## 1. Introduction

The period from birth to two years is a “critical window” for promoting optimal growth, health, and development, being mainly dependent on nutrition [1]. The World Health Organization (WHO) recommends exclusive breastfeeding until six months of age. Complementary feeding (CF), which involves introducing other foods and liquids alongside breast milk when breast milk alone can no longer meet the nutritional needs of infants, should begin at this time [2]. Specifically, the WHO recommends introducing complementary foods at six months of age [3]. In line with this, similar guidelines have been recommended by ESPGHAN, NASPGHAN, and the European Academy of Allergy and Clinical Immunology (EAACI), which advise introducing complementary foods between 17 and 26 weeks of age [4].

WHO identifies four indicators for appropriate CF: the timing of introduction, minimum meal frequency, dietary diversity, and acceptable diet. Failure to meet these criteria signifies inappropriate CF, while meeting them all indicates proper practice [5,6].

From around six months, a newborn’s nutritional needs increase, requiring CF to provide vital energy and nutrients necessary for growth and development. If complementary foods are introduced improperly or not at all, a child’s growth may be affected. The period from 6 to 23 months is a critical time for the onset of growth faltering, micronutrient deficiencies, and infectious diseases [7]. Additionally, there has been a rise in food allergies, particularly to eggs, shellfish, and nuts, in Western children. The early introduction of allergenic foods, such as oats, fish, and eggs, has been shown to reduce the risk of asthma, allergic rhinitis, and atopic dermatitis [8]. The introduction of gluten between 4 and 6 months may reduce the risk of celiac disease, with a gradual introduction being recommended while breastfeeding [9]. CF is vital for providing essential nutrients like iron, zinc, and vitamin A, with early inclusion of foods like meat, eggs, and liver being beneficial to address common deficiencies [10,11].

Malnutrition is characterized by an imbalance of energy and nutrients in the body, which negatively impacts the physical health of both children and adults [12]. One form of malnutrition, undernutrition, occurs when there is an insufficient intake of essential nutrients, including energy, high-quality protein, essential amino acids, vitamins, and minerals. This deficiency prevents the body from meeting its nutritional needs for proper growth, maintenance, and function [13]. Undernutrition accounts for almost one-third of all deaths in children under the age of five worldwide, with Asia accounting for the greatest percentage [14]. Approximately 21.3% of children under the age of five are stunted (too short for their height), 13% are underweight (too thin for their age), and 6.9% are wasting (low weight for their height) [15]. Malnutrition accounts for 60 percent of all under-five fatalities in underdeveloped nations [16]. Inadequate feeding habits, including a lack of dietary diversity, infrequent feeding, and improper timing of CF (either before 4 months or after 7 months), account for approximately 66% of all fatal accidents among children under five [17]. South Asia has a disproportionately high malnutrition burden compared to other areas, with the greatest rates of stunting (33.2%, about 60 million children) and wasting (14.8%, approximately 27 million children).

The Pakistan Demographic and Health Survey found that 45% of children under 5 were stunted, 11% were wasting, and 30% were underweight. The report also discovered that among babies aged 6 to 9 months, only 5.3% were solely breastfed, 10.3% received breast milk with water, 11% received breast milk with cow’s milk, and only 56.6% received CF in addition to breastmilk [18]. The existing literature on the determinants of child eating patterns, including minimal dietary diversity and meal frequency, showed that feeding practices are associated with the mother’s education, maternal profession, the child’s gender, and postnatal care [19]. Furthermore, the residence, family size, financial status, child age, the mother’s health (emotional and mental health) [20], and delivery site are important predictors of child feeding behaviors [21,22,23]. Recent research has also indicated that the birth interval, mother’s independence, and exposure to media are important determinants of dietary diversification [24].

Nutrition-related research in Pakistan and other low and middle-income countries generally focuses on improving breastfeeding patterns, with minimal focus on improving CF practices. However, one notable exception is the work by Muzi et. at. [25] which employed multi-level regressions to analyze the factors influencing CF practices in Pakistan, using data from the 2012–2013 Demographic and Health Survey. However, their analysis relies on data that are now eight years old, so the objective of this study is to provide fresh evidence of the knowledge of child-feeding practices in Pakistan and examine the relationship between child-feeding practices and individual-, household-, and community-level indicators using the most recent data.

This study is expected to provide valuable insights into the factors that influence early CF practices among children aged 9–23 months in Punjab, Pakistan. The anticipated results aim to identify key predictors of inappropriate feeding, including maternal education, employment, and health status, as well as socio-economic factors such as household income and family size. It is expected that the study will reveal a significant association between these factors and the rates of timely CF, meal frequency, dietary diversity, and acceptable diets. These findings are expected to underscore the importance of targeted interventions and education programs to improve maternal knowledge and promote healthier feeding behaviors, ultimately contributing to better child health outcomes and supporting economic growth in Pakistan. The results will also provide a foundation for future research and policy recommendations focused on enhancing child nutrition and health in the region.

## 2. Methodology

### 2.1. Study Design and Setting

This cross-sectional study examined the prevalence of inappropriate CF among infants aged 9–23 months. The study was conducted in multiple hospitals located in multiple cities across Pakistan, including both public and private sector healthcare facilities. The selected cities, namely Lahore, Multan, Islamabad, Gujranwala, and Faisalabad, were randomly chosen to ensure the representation of diverse geographical regions and populations within the country. The reason for choosing the hospital setting rather than a community setting was to ensure a diverse participant pool, as hospitals provide access to caregivers from various socio-economic backgrounds, including both urban and rural populations. Hospitals are frequently visited for routine check-ups and vaccinations, facilitating easier participant’s recruitment. Additionally, hospitals offer better infrastructure for data collection, such as access to medical records, which enhances the reliability of participant selection.

### 2.2. Study Participants and Eligibility Criteria

The target population for this study were infants between 9 and 23 months and their caregivers, primarily mothers, while the study population were only infants of 9–23 months. We mainly targeted this age group because appropriate CF is crucial during this developmental stage. Only those mothers who could read and write properly and signed the consent form were selected. Infants with congenital disorders or diseases that could potentially impact their diet, such as heart defects or autism, were excluded from the study.

### 2.3. Sample Size and Sampling Technique

The sample size for this study was determined using Cochran’s formula, resulting in a sample size of 600 participants. For the sampling technique, as the study was conducted in a hospital setting, a clinical survey approach was employed. Convenient sampling was used to select the hospitals, and participants were selected based on convenience, ensuring ease of access and recruitment from the available patient pool.

### 2.4. Instrument

The data collection for the study used structured and semi-structured questionnaire, consisting of both close-ended (multiple choice) and open-ended questions. It was organized into eight main sections with sub-questions focusing on gathering information from mothers regarding inappropriate CF. The first section collected demographic information about the child, including age, weight, height, gender, and birth order.

The second section of the questionnaire gathered parents’ information, including age, weight, education level, and employment status. The third section focused on household details, such as the number of children, place of delivery, monthly income, and healthcare access. The fourth section assessed mothers’ knowledge of CF through 12 questions, with responses of “Yes”, “No”, or “Don’t know”. Scores ranged from 0 to 12, categorizing knowledge as low (<4), moderate (4–8), or high (>9).

The fifth section addressed the child’s feeding history, including breastfeeding duration and the introduction of complementary foods. Section six covered the infant’s health status, such as gastrointestinal and respiratory issues. Section seven focused on access to clean water and hygiene practices, while section eight explored maternal health during various phases. The final section used a Food Frequency Questionnaire (FFQ) to assess dietary diversity, categorizing infants’ food intake based on eight food groups. A score of 1 was assigned for consuming four or fewer food groups, and 0 for more than four, helping evaluate the infant’s nutrition and dietary variety.

### 2.5. Data Collection

The researcher underwent training to conduct interviews and surveys with caregivers at various hospitals to collect data on CF practices and knowledge. Topics covered in the training included appropriate food types, feeding frequency, portion sizes, and hygiene. Data collection involved initial online Google forms and face-to-face interviews with mothers based on the form responses. Mothers who agreed to participate signed a form, and this process continued until the desired number of participants was achieved, ensuring all participants’ voluntary involvement in the study.

### 2.6. Operational Definitions

Timely introduction of complementary foods: It is recommended to start between 4 and 6 months. Mothers were asked about the introduction of solid foods to their children. Based on their responses, the introduction of complementary food was categorized into early (<4 months), timely (4–6 months), and late (>7 months) stages of CF. This categorization helped us to assess if the introduction of solid foods is aligned with the recommended guidelines.

Minimum dietary diversity: Minimum dietary diversity refers to the percentage of children aged 6–23 months who have consumed at least four of the eight food groups. There are eight dietary groups: breast milk, cereals, roots, and tubers, legumes and nuts, dairy products (infant formula, milk, yogurt, and cheese), flesh foods (meat, fish, and organ meats), eggs, vitamin A-rich fruits and vegetables, and other fruits and vegetables.

Dietary variety scores range from 0 to 7, where a score of zero means the child did not eat from any food group and a score of 7 indicates they ate from all food groups. We applied the minimum dietary diversity measure based on WHO recommendation. If a child consumed foods from four or more food categories, we gave them a score of “1” for minimum dietary variety. If they ate from less than four food groups, they received a score of “0”, indicating low dietary diversity.

Minimum meal frequency: The minimum meal frequency is defined as how many times a child receives complementary food in a day. According to WHO, it is twice for 6–8 months, three times for 9–11 months, three times for breastfed children aged 12–23 months, and four times for non-breastfed children. Appropriate feeding frequency was categorized as 1 (minimum meal frequency), whereas incorrect was recorded as 0 (low meal frequency).

Minimum acceptable diet: This is the combination of both the minimum dietary diversity and meal frequency.

### 2.7. Inappropriate Complementary Feeding Practice

CF methods that fail to meet the requirements for either a timely introduction or the minimum acceptable diet were assessed.

### 2.8. Statistical Analysis

Descriptive data of children, parents, and households were presented as frequencies and percentages. In addition, a chi-square test was used to determine the association between categorical variables, while a *t*-test was applied for continuous variables. Associations between inappropriate CF practices and other characteristics was examined using logistic regression. WHO guidelines were used to evaluate appropriate meal frequency, diversity, and acceptability. Dietary diversity was classified into seven WHO food groups to calculate the minimum nutritional variety. The odds ratios (ORs) were reported with 95% confidence intervals (CIs). A *p*-value of <0.05 was considered statistically significant. The SPSS version (26.0, IBM, Armonk, NY, USA) was used to conduct all the analyses.

## 3. Results

### 3.1. Socio-Demographic Characteristics of Participants

Among the demographic characteristics assessed in the study, a notable proportion of children fell within the age range of 13–18 months, comprising 45.4% of the total sample. This age range had the highest number of participants indicating a critical developmental period for infants. Regarding gender distribution, slightly more than half of the children (51%) were female. When examining body height, most children (57.5%) measured between 69 and 79 cm, indicating a relatively even distribution within this height range.

Moreover, the Body Mass Index (BMI) distribution revealed that a significant portion of children (45.5%) fell within the healthy weight category, while a smaller proportion were classified as underweight (18.2%), overweight (8.3%), or obese (2.5%). These findings emphasize the importance of monitoring growth parameters and nutritional status in early childhood, with implications for interventions promoting healthy development and preventing adverse health outcomes.

### 3.2. Complementary Feeding Indicators

This study reveals that a significant proportion (47.0%) of caregivers introduced CF before the recommended age of 4 months, compromising the health of infants. However, 39.7% adhered to guidelines by initiating CF between 4 and 6 months, which may improve infant health outcomes. Conversely, 30.9% started feeding after 7 months.

According to the results, 196 infants (32.7%) did not fulfill the WHO guidelines for CF instead of consuming the recommended 3–4 feeds appropriate for their age, these infants exhibited varied feeding patterns, including once a day, 12 times a day, or more than 4 times a day. Conversely, 67.2% of infants aged 9–23 months received appropriate CF, meeting this age group’s recommended frequency of 3–4 times daily. The prevalence of inappropriate complementary feeding and its key indicators are shown in Figure 1.

The study assessed dietary diversity among children aged 9–23 months across various food groups. The results indicate varying levels of dietary diversity across age groups. Among infants aged 9–13 months, wheat (18.2%), fruits (18.2%), and vegetables (33.4%) were the most commonly consumed food groups, while dairy products (44.5%) and rice (41.5%) were also prevalent. In the 14–18 months’ age group, fruits (44.5%) and dairy products (54.9%) exhibited higher consumption rates, with rice (33.2%) and white meat (35.5%) also significant. In the oldest age group (19–23 months), fruits (36.9%) and white meat (45.8%) remained popular choices, along with dairy products (51.4%) and vegetables (58.9%), indicating a broader dietary diversity among older infants.

The study found that 48.5% of the sample met the Minimum Acceptable Diet (MAD) criteria. This indicates that nearly half of the infants aged 6–23 months received a diet meeting the minimum standards for adequate nutrition.

### 3.3. Monovariate Factor Analysis of Inappropriate Complementary Feeding

The monovariate factor analysis of inappropriate complementary feeding in children, as presented in Table 1, reveals several significant associations between various demographic and socioeconomic factors and the practice of CF. Firstly, maternal employment status was significantly associated with CF practices (χ^2^ = 8.423, *p* = 0.038), with employed mothers exhibiting a lower prevalence of incomplete feeding compared to unemployed or retired mothers. Similarly, the father’s education level showed a significant association with CF practices (χ^2^ = 12.07, *p* = 0.017), with higher education levels correlating with lower rates of incomplete feeding.

Bivariate factor analysis of Inappropriate Complementary feeding

Furthermore, the number of children in the household was strongly associated with CF practices (χ^2^ = 22.64, *p* < 0.001), with households having more children showing higher rates of incomplete feeding. Additionally, average monthly income was significantly associated with CF practices (χ^2^ = 9.809, *p* = 0.007), with lower-income households exhibiting a higher prevalence of incomplete feeding.

Maternal knowledge scores regarding infant feeding practices also showed a significant association with CF practices (χ^2^ = 8.571, *p* = 0.014), with higher knowledge scores correlating with lower rates of incomplete feeding. The age of the child when CF was initiated also emerged as a significant factor (χ^2^ = 150.3, *p* < 0.001), with earlier initiation associated with higher rates of incomplete feeding.

Furthermore, the daily feeding frequency was significantly associated with CF practices (χ^2^ = 13.45, *p* = 0.004), with higher-frequency feeding correlating with lower rates of incomplete feeding. Lastly, emotional changes experienced by mothers when starting CF showed a significant association with CF practices (χ^2^ = 6.498, *p* = 0.039), with increased stress levels linked to higher rates of incomplete feeding.

Emotional changes during CF initiation were significantly associated with maternal stress levels. Mothers experiencing increased stress had a higher prevalence of incomplete feeding, with 22.3% reporting elevated stress levels compared to 13.8% who felt more emotionally stable and 27.2% whose emotional state remained unchanged.

In this study, several factors were found to be non-significant considering inappropriate CF practices. Demographic characteristics such as the age and gender of the children, as well as parental factors including maternal age, education level, and body characteristics (height and weight), did not show significant associations with CF practices. Similarly, household characteristics such as access to clean water, facilities for cooking clean food, and difficulty in accessing healthcare facilities were not significantly associated with inappropriate CF. Additionally, factors related to child-feeding practices such as breastfeeding initiation, frequency, and duration did not demonstrate significant associations with inappropriate CF practices.

To promote optimal infant feeding practices, socioeconomic, demographic, and emotional factors must be addressed, as these factors influence CF practices in multiple ways.

The logistic regression analysis provided insights into factors associated with inappropriate CF practices in infants (Table 2). Several demographic and socioeconomic variables showed significant associations. Infants born first in the birth order displayed notably higher odds of inappropriate CF (OR: 3.118, *p* = 0.048) compared to those born later in the birth order. Maternal employment status also appeared to be a significant factor, with unemployed mothers showing lower odds of inappropriate CF than employed counterparts (OR: 16.875, *p* = 0.015).

Moreover, certain parental characteristics and household dynamics were linked to inappropriate CF practices. For instance, fathers with postgraduate education displayed decreased odds of inappropriate CF (OR: 0.491, *p* = 0.002) compared to those with university-level education. Additionally, households with two children exhibited significantly lower odds of inappropriate CF (OR: 0.194, *p* = 0.008) than those with five or more children. Furthermore, financial factors played a role, with households earning less than 50k PKR per month having higher odds of inappropriate CF (OR: 1.879, *p* = 0.002) than those earning above 100k PKR per month.

Health-related factors also showed associations with inappropriate CF. For instance, infants introduced to CF between 4 and 6 months had substantially higher odds of inappropriate CF (OR: 13.91, *p* < 0.001) than those introduced after 7 months. Additionally, infants fed three to four times daily had higher odds of inappropriate CF (OR: 3.021, *p* = 0.038) compared to those fed four or more times daily.

The findings emphasize the complicated nature of factors influencing CF practices in infants, highlighting the need for targeted interventions that address socioeconomic, demographic, and health factors.

## 4. Discussion

This study aims to assess the prevalence of inappropriate CF among infants in Pakistan and identify factors associated with it while examining its impact on health outcomes. It is recommended to introduce CF at six months of age, as feeding before or after six months may lead to adverse health consequences.

In this study, questionnaire data were used to examine minimum dietary diversity, meal frequency, and acceptable diet among Pakistani children aged 9 to 23 months. However, according to this study, the practices of timely initiation of CF, minimum meal frequency, minimum dietary diversity, and minimum acceptable diet were 60.3%, 32.7%, 24.6%, and 48.5%, respectively, among mothers of children aged 9–23 months. The rate of timely CF initiation in our study is lower compared to regional countries, where 71% and 70% of children in Bangladesh [26] and Nepal [27]. However, Sri Lanka has the highest CF rate in South Asia, i.e., 84% [28]. The proportion of 6–23-month-old children who met the Minimum Meal Frequency (MMF) criteria in the current study was 32.7%. This is lower than the findings from Sri Lanka (88.3%), Bangladesh (81%), Nepal (82%), coastal South India (77.5%), Derashe, Southern Ethiopia (95%), and the Amibara district, Northeast Ethiopia (69.2%) [29,30,31,32,33,34]. However, it is nearly comparable to the results from Nagle Arsi (67.3%) and the Bale Zone, Ethiopia (68.4%) [35,36].

The disparity in MMF proportions among these studies may be attributed to differences in caregivers’ sociocultural, educational, and employment conditions. The current study found that 24.6% of children met the Minimum Dietary Diversity (MDD), which indicates that these children were fed from at least four out of the seven food groups: grains, roots and tubers; legumes and nuts; dairy products; flesh foods; vitamin A-rich foods; eggs; and other fruits and vegetables. When comparing this figure to other studies, the percentage in the current study is somewhat similar to Bangladesh, where 42% of children met the MDD. However, it is higher than the figures reported from the Nagle Arsi region (18.8%), Damota Sore (16%), India (15%), and Nepal (34%). On the other hand, the percentage is significantly lower than in Sri Lanka, where 71% of children met the MDD [37,38,39].

This study shows that the significant factors of inappropriate CF are the order of birth, mother’s employment status, parental education, number of children, household income, maternal knowledge, and maternal health. The findings show that maternal education, employment, and health significantly influence CF.

The work situation of mothers significantly influences the appropriateness of CF. Mothers working part-time have a reduced risk of adopting inappropriate CF compared to non-working mothers. This finding is consistent with a study conducted in Nepal [40], where working mothers, due to their increased empowerment and decision-making role in the household, are better able to manage their child’s feeding. On the other hand, in developed countries, working mothers often initiate CF before 6 months, which is considered inappropriate. This early initiation is linked to their work schedules and the belief that breastfeeding is outdated [41]. In Ethiopia, some studies have shown that working mothers may wean breastfeeding too early to return to work, while non-working mothers are more likely to start CF at the recommended 6 months. Various studies have revealed that a lack of knowledge of the mother was the primary reason for the early introduction of CF [42].

Our study shows a positive correlation between mothers’ education and inappropriate CF practices. Mothers with limited knowledge are less likely to know about CF, which aligns with findings from other studies [43,44,45]. Memon et al. described a positive relationship between maternal education and nutritional status, noting that some uneducated mothers delayed CF until their child was 1 year old [46]. Our qualitative research findings indicate that a significant number of mothers lack awareness regarding the appropriate age to initiate CF for their children. In their explanations, they cited various reasons for delaying the introduction of CF. Some expressed concerns that semi-solid and solid foods were too harsh for their children’s delicate digestive systems, fearing potential adverse reactions such as vomiting or diarrhea. Despite being the primary caregivers, many mothers reported that decisions regarding feeding practices, including the introduction of CF, are often made by their husbands and mothers-in-law. Additionally, mothers acknowledged that their mothers-in-law possess greater knowledge about the nutritional requirements of children and the importance of providing them with nutrient-rich foods.

Regarding household characteristics, the monthly income also positively showed an association with inappropriate CF, similar to a published study, which showed that the percentage of timely CF at 6–8 months was higher in richer and richest households, at 51.3 and 70.9%, respectively, compared with poorest, poorer, and middle households, which achieved rates of 28.0, 33.7, and 23.1%, respectively [47]. This association are in accordance with other studies [3,28,47,48], where they also highlighted how women perceived poverty as a barrier to adequately feeding their children [48], while Liaqat et al. identified poverty as a primary factor contributing to inappropriate feeding practices [49].

In Pakistan, the timing of CF is a significant challenge, as the percentage of timely introductions of complementary food is lower than in some other South Asian nations. One of the studies conducted by Shamim et al. [50] indicates that the premature initiation of CF has been linked to an elevated risk of infections, and study [51] showed that a delayed introduction of CF negatively impacts growth and may contribute to problematic eating behaviors such as food rejection and difficulty in learning to chew. Late or early intake of complementary foods in Pakistani children is a critical concern, as a diverse diet is connected to an optimal micronutrient intake and a decreased risk of stunted development among children in underdeveloped countries.

Unlike previous research in Pakistan and other low- and middle-income countries that has primarily focused on breastfeeding, our study uniquely emphasized improving CF practices. While other study [25] like that of analyzed CF practices using data from the 2012–2013 Demographic and Health Survey, our research provides fresh, up-to-date evidence on child-feeding behaviors. Our findings reveal that factors such as childbirth order, the mother’s employment, parental education, number of children, income, maternal knowledge, and maternal health significantly influence CF practices. Specifically, maternal education, employment, and health were found to have a strong impact on appropriate feeding behaviors. This highlights the importance of these socio-economic and maternal factors in improving child nutritional outcomes. Our study fills existing gaps in the literature and offers valuable insights for targeted interventions and policy recommendations to enhance CF practices in Pakistan.

### Limitations of the Study

The study has several limitations that need to be considered. First, the cross-sectional nature of the data limits our ability to draw causal conclusions. Additionally, the questionnaire was administered only once, which may introduce respondent biases, such as socially desirable answers. The survey did not include detailed information on the quantity and quality of food consumed, making it difficult to assess nutritional adequacy. Furthermore, the data were collected in a hospital setting, which may not be fully representative of the general population. This setting requires participants to have access to Google and literacy skills, limiting the sample to those with these resources. A community center could have provided a broader and more diverse sample.

## 5. Conclusions

Our research found that the inappropriate CF practices of infants are strongly related to maternal education, job, number of children, maternal knowledge, birth order, and the mother’s health. As a result, efforts must be made in laws and regulations to promote safe breastfeeding and CF, female education, and the availability of inexpensive, nutritious, and diversified meals.

## Figures and Tables

**Figure 1 nutrients-16-04379-f001:**
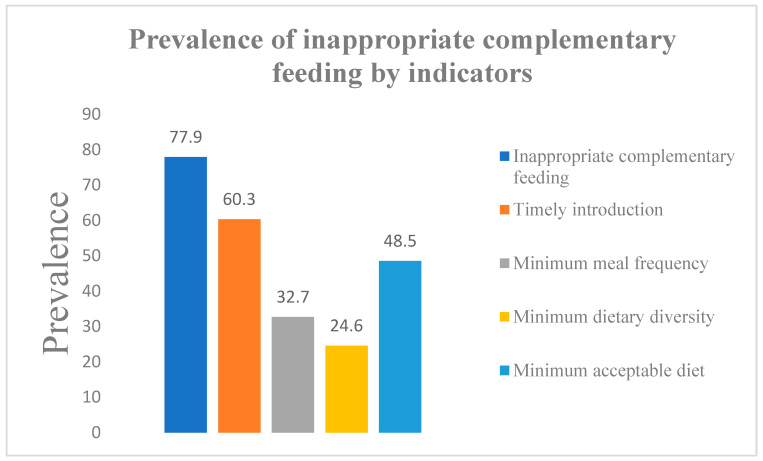
Prevalence of inappropriate complementary feeding by indicators.

**Table 1 nutrients-16-04379-t001:** Summarized demographic characteristics and monovariate factor analysis of inappropriate complementary feeding in children.

Variables	N (%)	Inappropriate Complementary Feeding		Test Statistics		*p* Value
		Yes (%)	No (%)			
**Age (month)**				1.725		0.42
7–9	109 (18.2%)	28 (25.7)	81 (74.3)			
10–18	273 (45.4%)	61 (22.3)	212 (77.7)			
19–23	218 (36.3%)	42 (19.3)	176 (80.7)			
**Gender**				0.128		0.72
Male	294 (49%)	66 (22.4)	228 (77.6)			
Female	306 (51%)	65 (21.2)	241 (78.8)			
**Body height (cm)**				1.799		0.40
Mean ± SD 54–68 cm	75.7 ± 9.4 101 (16.8%)	76.1 ± 9.8 26 (25.7)	75.6 ± 9.2 75 (74.3)	0.523		0.60
69–79 cm	345 (57.5%)	69 (20)	276 (80)			
89 and above	154 (25.7%)	36 (23.4)	118 (76.6)			
**Body weight (kg)**				1.725		0.422
Mean ± SD 7–9 kg	11.8 ± 2.6 314 (52.3%)	11.7 ± 2.7 68 (21.7)	11.9 ± 2.6 246 (78.3)	0.756		0.451
10–12 kg	125 (20.8%)	23 (18.4)	102 (81.6)			
9–23 kg	161 (26.8%)	40 (24.8)	121 (75.2)			
**BMI (kg/m^2^)**				4.718		0.194
Mean ± SD Less than 18.5 (underweight)	20.8 ± 3.6 140 (18.2%)	20.7 ± 3.3 26 (18.6)	20.9 ± 3.7 114 (81)	0.597		0.55
Btw 18.5–24.9 (healthy weight)	395 (45.5%)	96 (24.3)	299 (75.7)			
Among 25–29.9 (overweight)	50 (36.3%)	7 (14)	43 (86)			
30.0 or higher above (obese)	15 (2.5%)	2 (13.3)	13 (86.7)			
**Birth’s Order**				1.452		0.693
First-born	72 (12%)	16 (22.2)	56 (77.8)			
Second-born	225 (37.5%)	51 (22.7)	174 (77.3)			
Third-born	256 (42.7%)	57 (22.3)	199 (77.7)			
Fourth and above	47 (7.8%)	7 (14.9)	40 (85.1)			
** Parents Information **						
** Age of the mother ** **(year)**				7.006		0.072
20–30	341 (56.8%)	76 (26.3)	213 (73.7)			
31–40	213 (35.5%)	42 (18.6)	184 (81.4)			
41–50	40 (6.7%)	12 (15.8)	64 (84.4)			
51–60	6 (1%)	1 (11.1)	8 (88.9)			
**Height of the mother (cm)**				0.885		0.829
Mean ± SD 150 cm	165.6 ± 7.3 2 (0.3%)	164.2 ± 3.5 20 (22.2)	165.8 ± 6.7 70 (77.8)	0.231		0.82
152–165 cm	312 (52%)	65 (21.7)	235 (78.3)			
167–180 cm	285 (47.5%)	45 (22.5)	155 (77.5)			
182–193 cm	1 (0.2%)	1 (10)	9 (90)			
**Body weight (kg)**				0.252		0.969
Mean ± SD 60–70 kg	81.2 ± 10.3 99 (16.5%)	81.5 ± 9.9 20 (20.2)	81.1 ± 10.1 79 (79.8)	0.405		0.69
71–80	181 (30.2%)	40 (22.1)	141 (77.9)			
81–90	189 (31.5%)	41 (22.7)	148 (78.3)			
Above	131 (21.8%)	30 (22.9)	101 (77.1)			
**Mothers Education**				2.552		0.635
None	171 (28.5%)	38 (22.2)	133 (77.8)			
Secondary school	316 (52.7%)	73 (23.1)	243 (76.9)			
FSC	93 (15.5%)	15 (16.1)	78 (83.9)			
University Level	14 (2.5%)	3 (21.4)	11 (78.6			
Postgraduate	10 (1%)	2 (33.3)	4 (66.7)			
**Employment of Mother**				8.423		0.038 *
Employed	28 (4.7%)	1 (3.6)	27 (96.4)			
Not employed	13 (2.2%)	5 (38.5)	8 (61.5)			
Retired	408 (68%)	95 (23.3)	313 (76.7)			
Own business	151 (25.2)	30 (19.9)	121 (80)			
** Age of the Father ** **(year)**				2.611		0.625
20–30		9 (32.1)	19 (67.9)			
31–40		63 (21.6)	228 (78.4)			
41–50		32 (20)	127 (79.9)			
51–60		27 (22.5)	93 (77.5)			
Above 60		0 (0)	2 (100)			
**Father’s Education**				12.07		0.017 *
None		2 (50)	2 (50)			
Secondary school		0 (0)	1 (100)			
high school		28 (19.7)	114 (80.3)			
Undergraduate		58 (18.4)	258 (81.6)			
Graduation		43 (31.4)	94 (68.6)			
**Employment of Father**				2.149		0.542
Employed		75 (22.3)	262 (77.7)			
Not employed		3 (16.7)	15 (83.3)			
Retired		1 (7.1)	13 (92.9)			
Own business		52 (22.5)	179 (77.5)			
** Household Information **						
** Number of Children **				22.64		<0.001 *
1 Child		0 (0)	1 (100)			
2 Child		6 (7.2)	77 (92.8)			
3 Child		54 (19.1)	229 (80.9)			
4 Child		62 (31.3)	136 (68.7)			
5 or above		9 (25.7)	26 (74.3)			
**Have enough food for children**				1.799		0.180
Yes		78 (20.2)	309 (79.8)			
No		53 (24.9)	160 (75)			
**Place of Delivery**				0.123		0.726
Home		46 (22.7)	157 (77.3)			
Health Institution		85 (21.4)	312 (78.3)			
**Sex of Household**				0.178		0.673
Male		127 (22)	451 (78)			
Female		4 (18.2)	18 (81.8)			
**Average Monthly Income**				9.809		0.007 *
<50k PKR		49 (16.5)	248 (83.5)			
50k–100k PKR		75 (27.1)	202 (72.9)			
>100k PKR		7 (26.9)	19 (73.1)			
**Difficulty in accessing healthcare facilities**				0.590		0.443
Yes		77 (23)	258 (77)			
No		54 (20.4)	211 (79.6)			
**The score of Maternal knowledge In complementary feeding practice**						
**Mean overall score**					8.571	0.014 *
<4		40 (16.4)	204 (83.)			
4–8		87 (26.3)	244 (73.3)			
9–12		4 (16)	21 (84)			
**Child Feeding**						
** Has your child been fed by their mother’s milk? **				1.489		0.223
Yes		112 (21.1)	419 (78.9)			
No		19 (27.5)	50 (72.5)			
**Still breastfeeding your child?**				1.062		0.588
Yes		85 (22.2)	298 (77.8)			
No		45 (20.9)	170 (79.1)			
**Age at which the child stops breastfeeding?**				** 0.984 **		** 0.805 **
<6 months		15 (22.4)	52 (77.6)			
6–12 months		34 (24.6)	104 (75.4)			
>12 months		54 (20.4)	211 (79.6)			
Skip		28 (21.5)	102 (78.50			
**Age of the child when complementary feeding was started?**				** 150.3 **		** <0.001 **
<4 months		23 (8.2)	259 (91)			
4–6 months		98 (52.7)	88 (47.3)			
>7 months		10 (7.6)	122 (92.4)			
**Reason for starting complementary food at the chosen age?**				** 0.627 **		** 0.890 **
Baby seemed interested in food		16 (21.1)	60 (78.9)			
Healthcare recommendation		36 (23.8)	115 (76.2)			
Family or cultural tradition		47 (20.5)	182 (79.5)			
I was unsure when to start		32 (22.2)	112 (77.8)			
**Number of times you fed your child per day**				** 13.45 **		** 0.004 **
Once only		19 (15.6)	103 (84.4)			
2–3 times		48 (19.2)	202 (80)			
3–4 times		58 (30.7)	131 (69.3)			
4+ times		6 (15.4)	33 (84.5)			
**Information on Clean Water**						
**Access to clean water**				** 1.245 **		** 0.264 **
Yes		16 (27.6)	42 (72.4)			
No		115 (21.2)	427 (78.8)			
** Does your domestic setup has the facilities to cook clean food? **				** 2.341 **		** 0.126 **
Yes		110 (20.9)	417 (79.1)			
No		21 (28.8)	52 (71.2)			
**Mothers Health**						
**Have you experienced any of these changes when started CF**				** 5.715 **		** 0.335 **
Weight gain		42 (89.4)	5 (10.6)			
Weight loss		47 (72.3)	18 (27.7)			
Digestive problems		85 (77.3)	25 (22.7)			
Fatigue		112 (80.6)	27 (19.4)			
Emotional changes		105 (77.2)	31 (22.8)			
Skin changes		78 (75.7)	25 (24.3)			
**Have you experienced any emotional changes when starting CF**				** 6.498 **		** 0.039 **
Felt more emotionally stable		94 (86.2)	15 (13.8)			
Yes, my stress increases		276 (77.7)	79 (22.3)			
No Its same		99 (72.8)	37 (27.2)			
**Have you experienced any physical changes when starting CF**				** 5.084 **		** 0.079 **
I have noticed Positive changes		49 (27.7)	128 (72.3)			
I have noticed negative changes		49 (19.8)	199 (80.2)			
No its same		33 (18.9)	142 (81.1)			

* The value is statistically significat at *p* < 0.05.

**Table 2 nutrients-16-04379-t002:** The multivariate factor analysis of inappropriate complementary feeding in children.

Variables	OR (95%CI)	*p*-Value
**Body weight (kg)**		
7–9	1.451 (0.56–3.7)	0.44
10–18	1.094 (0.6–1.8)	0.74
19–23 (ref)		
**Gender**		
Male	1.197 (0.7–1.8)	0.419
Female (ref)		
**Body height** (cm)		
54–68	0.765 (0.2–2.5)	0.661
69–79	0.668 (0.3–1.3)	0.250
89 and Above (ref)		
**Body weight (kg)**		
7–9	0.784 (0.3–1.5)	0.493
10–12	0.791 (0.4–1.5)	0.488
9–23 (ref)		
**BMI (kg/m^2^)**		
Less than 18.5 (underweight)	0.936 (1.4–5.9)	0.944
Btw 18.5–24.9 (healthy weight)	1.496 (0.2–8.3)	0.647
Among 25–29.9 (overweight)	0.875 (0.13–5.6)	0.889
30.0 or higher above (obese) (ref)		
**Birth’s order**		
First-born	3.118 (1.0–9.6)	0.048 *
Second-born	2.537 (0.9–6.7)	0.063
Third-born	1.954 (0.7–5.07)	0.169
Fourth-born (ref)		
** Parents Information **		
** Age of the mother (year) **		
20–30	2.854 (0.3–23)	0.327
31–40	1.826 (0.2–14)	0.575
41–50	1.500 (1.7–13)	0.714
51–60 (ref)		
**Height of the mother** (cm)		
150	2.571 (0.3–21)	0.384
152–165	2.489 (0.3–20)	0.391
167–180	2.613 (0.3–21)	0.368
182–193 (ref)		
**Body weight (kg)**		
60–70	0.680 (0.3–1.4)	0.299
71–80	0.966 (0.5–1.7)	0.914
81–90	1.052 (0.5–1.9)	0.870
Above (ref)		
**Mothers Education**		
None	0.589 (0.1–3.4)	0.556
Metric	0.624 (0.1–3.5)	0.595
FSC	0.388 (0.06–2.3)	0.308
University Level	0.535 (0.06–4.6)	0.569
Postgraduate (ref)		
**Employment of mother**		
Unemployed	16.875 (1.7–166)	0.015 *
Retired	8.195 (1–61)	0.040 *
Self employed	6.694 (0.8–51)	0.067
Employed (ref)		
**Age of the Father (year)**		
20–30	0.987 (0.5–1.8)	0.968
31–40	1.067 (0.5–2.0)	0.842
41–50	0.910 (0.4–1.9)	0.804
Above 60 (ref)		
**Father’s Education**		
None	2.186 (0.2–16)	0.442
FSC	0.537 (0.3–0.9)	0.026 *
Postgraduate	0.491 (0.3–0.77)	0.002 *
University Level (ref)		
**Employment of Father**		
Employed	1.143 (0.7–1.7)	0.562
Not employed	0.986 (0.2–4.2)	0.985
Retired	0.245 (0.02–2.1)	0.208
Self-employed (ref)		
** Household Information **		
** Number of Children **		
2	0.194 (0.50.6)	0.008 *
3	0.621 (0.2–1.5)	0.310
4	1.249 (0.5–3.1)	0.633
5 or above (ref)		
**Have enough food for children**		
Yes	0.720 (0.4–1.1)	0.147
No (ref)		
**Place of Delivery**		
Home	1.020 (0.6–1.6)	0.931
Health Institution (ref)		
**Sex of Household**		
Male	0.832 (0.2–2.8)	0.772
Female (ref)		
**Average Monthly Income** (×1000 PKR)		
<50	1.879 (1.2–2.8)	0.002 *
50–100	1.865 (0.7–4.6)	0.184
>100 (ref)		
**Difficulty in accessing healthcare facilities**		
Yes	1.080 (0.6–1.6)	0.735
No (ref)		
**Maternal knowledge**		
<4	1.818 (1.1–2.7)	0.005 *
4–8	0.971 (0.3–2.9)	0.960
9–12 (ref)		
**Child Feeding**		
** Has your child been fed by their mother’s milk? **		
Yes	0.639 (0.3–1.2)	0.177
No (ref)		
**Still breastfeeding your child?**		
Yes	0.443 (0.02–7.6)	0.576
No	0.360 (0.02–6)	0.476
**Age at which the child stops breastfeeding?** (Months)		
<6	1.051 (0.5–2.1)	0.891
6–12	1.191 (0.6–2.1)	0.548
>12	0.932 (0.5–1.5)	0.789
Skip (ref)		
**Age of the child when complementary feeding was started?** (Months)		
<4	1.041 (0.4–2.2)	0.919
4–6	13.91 (6.8–28.4)	<0.001 *
>7 (ref)		
**Reason for starting complementary food at the chosen age?**		
Baby seemed interested in food	0.964 (0.4–1.9)	0.916
Healthcare recommendation	1.117 (0.6–1.9)	0.693
Family or cultural tradition	0.917 (0.5–1.5)	0.740
I was unsure when to start (ref)		
**Number of times you fed your child per day**		
Once only	1.093 (0.3–3.3)	0.876
2–3 times	1.426 (0.5–4.0)	0.503
3–4 times	3.021 (1.0–8.5)	0.038 *
4+ times (ref)		
**Information about clean water**		
**Access to clean water**		
Yes	0.1443 (0.7–2.6)	0.241
No (ref)		
** Does your domestic setup have the facilities to cook clean food? **		
Yes	0.644 (0.3–1.11)	0.117
No (ref)		
** Mothers Health **		
**Have you experienced any of these changes when started CF**		
Weight gain	0.371 (1.1–1.04)	0.061
Weight loss	1.285 (0.6–2.5)	0.494
Digestive problems	0.968 (0.5–1.8)	0.922
Fatigue	0.750 (0.4–1.3)	0.365
Emotional changes	0.962 (0.5–1.7)	0.902
Skin changes (ref)		
**Have you experienced any emotional changes when starting CF**		
Felt more emotionally stable	1.679 (1–2.7)	0.047 *
Yes, my stress increases	1.024 (0.6–1.6)	0.925
It’s the same (ref)		
**Have you experienced any emotional changes when starting CF**		
I have noticed positive changes	0.404 (0.2–0.7)	0.008 *
I have noticed negative changes	0.747 (0.4–1.1)	0.214
It’s the same (ref)		

* The value is statistically significat at *p* < 0.05.

## Data Availability

Complementary feeding and its associated factors among infants’ data will be made available on demand.

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
