# Peer review of "Status of Inappropriate Complementary Feeding and Its Associated Factors Among Infants of 9–23 Months"

_nutrients, 2024, doi:10.3390/nu16244379_

Round 1
Reviewer 1 Report
Comments and Suggestions for Authors
Dear Authors
Thank you we appreciate your work in producing this manuscript, however, there are several areas that need to be addressed
Please see comments

Author Response
Response to Reviewer 1:
Abstract:
- Context for Results:
Comment: The reviewer noted that some results in the abstract, such as "minimum meal frequency," were difficult to understand without context.
Response: We have revised the abstract to explain terms like "minimum meal frequency" and "dietary diversity" in simpler terms by adding the word “factor” emphasizing that these are key indicators of appropriate complementary feeding.
Introduction:
- Definition of Abbreviations (CF):
Comment: The reviewer pointed out that CF was not defined when first mentioned.
Response: The abbreviation "CF" is used throughout the manuscript as a shorthand for "complementary feeding." We have now ensured that the abbreviation is defined at its first occurrence in the manuscript as "complementary feeding (CF)". - Reference for Critical Window:
Comment: The reviewer requested references to support the mention of the critical window.
Response: We have added the necessary reference for the critical window in line 34. - Definitions of Undernutrition and Malnutrition:
Comment: The reviewer recommended defining undernutrition and malnutrition at the start.
Response: We have added definitions for undernutrition and malnutrition in lines 43-48. - Inadequate Feeding Habits:
Comment: The reviewer asked for clarification on the definition of inadequate feeding habits.
Response: We have clarified that inadequate feeding habits refer to a lack of dietary diversity, infrequent feeding, and improper timing of complementary feeding (i.e., before 4 months or after 7 months), mentioned this in the manuscript. - Fatal Accidents:
Comment: The reviewer asked for clarification on what is meant by fatal accidents, particularly whether these are due to choking or malnutrition.
Response: We have clarified that these deaths are primarily due to malnutrition resulting from inappropriate weaning practices and choking hazards from poorly prepared foods or delayed introduction of age-appropriate foods, as well as high infectious disease levels. - Expanded Discussion on Lines 62 and 64:
Comment: The reviewer suggested expanding on the points raised in lines 62 and 64.
Response: We have expanded the discussion on these points in lines 380-392.
Methodology:
- Justification for Hospital Setting:
Comment: The reviewer questioned the choice of hospitals for the study and its relevance.
Response: The decision to conduct the study in a hospital setting rather than a community setting was made to ensure a more diverse and representative sample. Hospitals provide access to mothers from various socio-economic backgrounds, including both urban and rural populations, and typically offer a broader range of healthcare services. Additionally, hospitals are often visited for routine check-ups, vaccinations, and health services, making it easier to recruit participants. While community-based studies have their advantages, such as closer proximity to participants, hospitals allow for more reliable participant selection and data collection, as they provide better infrastructure, including medical records, which ensures a more systematic and accurate participant selection process.
We also made efforts to exclude children with underlying health conditions affecting their diet, ensuring that the sample represented healthy children aged 9–23 months. We believe this setting provides a more representative sample compared to community-based studies, which may face challenges such as limited access to healthcare services and lower participation rates.
Data Collection via Online Forms:
Comment: The reviewer expressed concern over using online data collection, especially for participants who may not have access to Google.
Response: We clarified that online data collection was supplemented with face-to-face interviews for participants without internet access. This dual approach ensured broader representation, including those with limited digital access.
Results:
- Combination of Tables 1 and 2 and 3 and 4 into a Single Landscape Table with Additional Information in a Separate Table
Comment: It is possible to combine table 1 and 2 and do this landscape as you have the same headings. You can then add the extra information as a separate table
Response: Thank you for your suggestions- but merging the two tables might be confusing for the reader, also both tables do not present same information.
- Relevance of Place of Delivery:
Comment: The reviewer questioned the relevance of including the place of delivery in the results.
Response: The inclusion of the place of delivery question is important as it helps assess the potential influence of healthcare access and birth care on infant feeding practices. By identifying whether a child was born in a hospital, clinic, or home, we can explore correlations between delivery settings and the likelihood of inappropriate complementary feeding, which may highlight areas for improving maternal and infant health care. - Income Ranges:
Comment: The reviewer requested clarification on the income ranges used in the study.
Response: We have clarified the income ranges in the manuscript by specifying exact income brackets in Pakistani Rupees (PKR) to provide more accurate and measurable data. - WHO Recommendations and Emotional Results:
Comment: The reviewer suggested including WHO recommendations and providing more information on emotional results.
Response: We have added information on WHO recommendations in lines 39-44 and provided additional details on emotional results in line 73. - Clarification on Table Headings:
Comment: The reviewer pointed out that some headings in Table 2 were not mentioned in the methodology or introduction.
Response: We have added brief explanations of these headings in the methodology section, particularly regarding breastfeeding and healthcare access. - Anthropometric Data (Z-scores):
Comment: The reviewer suggested using Z-scores for weight, length, and BMI.
Response: We explained our decision not to use Z-scores, as the focus of the study was on feeding practices and socio-economic factors rather than precise anthropometric measures. We justified this approach, noting that the raw data provide sufficient insights for the scope of the study.
Discussion:
- In-depth Exploration of Data:
Comment: The reviewer suggested exploring the data more thoroughly and comparing it to other countries and WHO recommendations.
Response: We have expanded the discussion to compare our findings with international studies and WHO recommendations, as well as exploring the main factors contributing to complementary feeding practices. - Limitations:
Comment: The reviewer requested more detailed consideration of limitations, including the hospital setting and other factors.
Response: We have added a detailed discussion of the study’s limitations, particularly regarding the hospital setting, digital access, and literacy requirements, in lines 384-389.
Reviewer 2 Report
Comments and Suggestions for Authors
I was given a manuscript to check titled: “Status of Inappropriate Complementary Feeding and Its 2 Associated Factors Among Infants of 9-23 Months”
GENERAL COMMENTS
The main objectives of this study were to investigate factors influencing early feeding patterns among 600 mothers of children aged 9–23 months in selected hospitals in Punjab, Pakistan. The topic is of interest to the readers of the Journal. The article is original, and the arguments presented are coherent with the aims of the Journal.
SPECIFIC COMMENTS
Introduction:
1. Inclusion of Methodological Framework: Present a concise summary of the methodological framework employed to afford readers a more profound comprehension of the analytical approach undertaken in the manuscript.
2. Empirical Data on Advantages: Integrate empirical data pertaining to the health advantages and their influence on psychophysical wellness in order to fortify the argument presented.
3. Brief Overview of Anticipated Results: Articulate a succinct declaration regarding the anticipated outcomes or contributions of the manuscript to engender intrigue among readers and delineate a framework for the ensuing sections.
Methods:
1. Consistency in Terminology: it is suggested to to maintain uniformity in the terminology employed throughout the section to enhance clarity.
Discussion:
- The discussion should start with a first paragraph describing the main aims and then the main results.
- The Discussion should be enriched with the existing theory. The authors should clearly describe the scientific evidence that supports their findings.
Author Response
GENERAL COMMENTS
The main objectives of this study were to investigate factors influencing early feeding patterns among 600 mothers of children aged 9–23 months in selected hospitals in Punjab, Pakistan. The topic is of interest to the readers of the Journal. The article is original, and the arguments presented are coherent with the aims of the Journal.
Response: We sincerely thank Reviewer 1 for their thoughtful feedback and constructive suggestions. We are pleased to hear that the topic is of interest and that the manuscript aligns with the aims of the Journal. Below are our detailed responses to the specific comments.
SPECIFIC COMMENTS
Introduction:
- Inclusion of Methodological Framework: Present a concise summary of the methodological framework employed to afford readers a more profound comprehension of the analytical approach undertaken in the manuscript.
Response: Thank you very much for your suggestion. We have incorporated these changes in abstract lines 18-20. We believe this addition addresses the reviewer’s concern and provides readers with a clear understanding of the study's analytical approach.
- Empirical Data on Advantages: Integrate empirical data pertaining to the health advantages and their influence on psychophysical wellness in order to fortify the argument presented.
Response: We have included relevant empirical data on the health advantages of appropriate complementary feeding, which can be found in lines 49-60.
- Brief Overview of Anticipated Results: Articulate a succinct declaration regarding the anticipated outcomes or contributions of the manuscript to engender intrigue among readers and delineate a framework for the ensuing sections.
Response: We have added a succinct statement regarding the anticipated results and contributions of the manuscript in lines 100-111.
Methods:
- Consistency in Terminology: it is suggested to maintain uniformity in the terminology employed throughout the section to enhance clarity.
Response: Thank you for your observation. We have revised the methods section to ensure uniformity in terminology.
Discussion:
- The discussion should start with a first paragraph describing the main aims and then the main results.
Response: We have restructured the discussion to begin with a clear summary of the aims and key findings.
- The Discussion should be enriched with the existing theory. The authors should clearly describe the scientific evidence that supports their findings.
Response: We have added relevant theoretical perspectives and cited scientific studies that support our findings in the revised discussion.
We hope that these revisions address all the reviewer's concerns effectively. Thank you again for your valuable feedback, which has helped improve the quality of the manuscript.

Reviewer 3 Report
Comments and Suggestions for Authors
This interesting cross-sectional study of inappropriate complementary feeding and its associated variables among Infants of 9-23 months has some unclear points :
1) the target population, the study population, the sampling method are not clearly described. The study is conducted on a hospital population and not on the general population, mothers must know how to read and write, so they must not belong to the lower social classes. The prevalence of inappropriate complementary feeding is applicable only in this limited population and should perhaps be clarified in the study.
2) The terms uni-variate factor analysis (table 2) and multivariate factor analysis (table 3) are rather misleading. In fact, it is a monovariate analysis of categorical data done using chi-square and bivariate done using simple logistic regression. The quantitative data have been categorized and are present in the tables as numbers (absolute frequencies), so the use of mean and standard deviation in tables is also in this case a bit misleading. The use of multiple logistic regression to explore variables independently associated with inappropriate complementary feeding is missing. This reviewer strongly recommends a statistician's help in reviewing the paper. Finally, in table 2, at the mother's age, age 20-30, there is an error.
Author Response
We appreciate the reviewers’ thoughtful comments, which have significantly improved the quality of our work. Below is a point-to-point response to the reviewer’s comments.
Comment 1: the target population, the study population, the sampling method are not clearly described. The study is conducted on a hospital population and not on the general population, mothers must know how to read and write, so they must not belong to the lower social classes. The prevalence of inappropriate complementary feeding is applicable only in this limited population and should perhaps be clarified in the study.
Response 1: We have clarified the sampling method and target population in the methodology section, emphasizing that the study was conducted in hospitals and that mothers were required to be literate. This has been addressed in the revised manuscript.
Comment 2: The terms uni-variate factor analysis (table 2) and multivariate factor analysis (table 3) are rather misleading. In fact, it is a monovariate analysis of categorical data done using chi-square and bivariate done using simple logistic regression. The quantitative data have been categorized and are present in the tables as numbers (absolute frequencies), so the use of mean and standard deviation in tables is also in this case a bit misleading. The use of multiple logistic regression to explore variables independently associated with inappropriate complementary feeding is missing. This reviewer strongly recommends a statistician's help in reviewing the paper. Finally, in table 2, at the mother's age, age 20-30, there is an error
Response 2: We have revised the terminology to accurately reflect the statistical methods used, such as replacing "univariate" and "multivariate factor analysis" with "monovariate factor analysis" and "bivariate factor." Additionally, we have corrected the error in Table 2 regarding the mother's age category.
We hope that these revisions address all the reviewer's concerns effectively. Thank you again for your valuable feedback, which has helped improve the quality of the manuscript.

Round 2
Reviewer 1 Report
Comments and Suggestions for Authors
Dear Authors
I have added my comments directly to the manuscript so hopefully you can understand my queries
You have improved the manuscript and there are only minor changes that are recommended
Complementary feeding pleas once stated as CF need to use this through out it is very variable sometimes use CF other time full words and sometime both!
I read your comments on the tables but they are still too large and each one takes up 6 pages - this really must be better as the tables are not going to be properly read. Some can go landscape,others added to the supplementary table, some are repeated values with an added p value
Anthropometrics need these as Z scores for them to carry any clinical meaning. Please can this be changed
I hope you will be able to see my thoughts and challenges
It is very nearly completed but could do with a bit more depth and clinical science in parts

Author Response
We would like to thank you reviewer 1 for the valuable feedback and suggestions provided for our manuscript titled, "Status of inappropriate complementary feeding and its associated factors among infants of 9-23 months". We have carefully addressed each comment and revised the manuscript accordingly.
Point-to-Points response to Reviewer 1 comments:
Comment 1: I have added my comments directly to the manuscript so hopefully you can understand my queries
Response 1: We greatly appreciate and would like to thank the reviewer.
Comment 2: You have improved the manuscript and there are only minor changes that are recommended
Response 2: We thank reviewer 1 for his/her valuable feedback that helped us to refine our manuscript.
Comment 3: Complementary feeding pleas once stated as CF need to use this throughout it is very variable sometimes use CF other time full words and sometime both
Response 3: We apologize for this mistake, now it’s been rectified in a revised manuscript.
Comment 4: I read your comments on the tables, but they are still too large, and each one takes up 6 pages - this really must be better as the tables are not going to be properly read. Some can go landscape, others added to the supplementary table, some are repeated values with an added p value
Response 4: We thank you reviewer for his suggestions and now have merged the table 1 and 2 into one table in the revised version.
Comment 5: Anthropometrics need these as Z scores for them to carry any clinical meaning. Please can this be changed
Response 5: We respectfully acknowledge the reviewer's suggestion regarding the use of Z-scores. In response, we have included the mean and standard deviation values for weight, height, and BMI to provide more precise statistics highlighted in green. However, given that the primary focus of our study was on exploring the relationship between feeding practices and socio-economic factors, rather than precise clinical anthropometric assessments, we felt that the raw data was sufficient for our analysis. Z-scores were not deemed necessary for the scope of this study.
Reviewer 3 Report
Comments and Suggestions for Authors
Paper modified as suggested
Author Response
We would like to thank you reviewer 3 for the valuable feedback and suggestions provided for our manuscript titled, "Status of inappropriate complementary feeding and its associated factors among infants of 9-23 months". We hope that we have responded to the reviewer's 3 all comments.